



**Exploring wintertime regional haze in Northeast China: role**
**of coal and biomass burning**
**Jian Zhang[1], Lei Liu[1], Liang Xu[1], Qiuhan Lin[1], Hujia Zhao[2], Zhibin Wang[3],**
**Song Guo[4], Min Hu[4], Zongbo Shi[5], Dantong Liu[1], Dao Huang[1], and Weijun Li[1] [*]**
[1]Department of Atmospheric Sciences, School of Earth Sciences, Zhejiang University,
Hangzhou, 310027, China
[2]Institute of Atmospheric Environment, China Meteorological Administration,
Shenyang, 110016, China
[3]Research Center for Air Pollution and Health, College of Environmental and
Resource Sciences, Zhejiang University, Hangzhou, 310058, China
[4]State Key Joint Laboratory of Environmental Simulation and Pollution Control,
College of Environmental Sciences and Engineering, Peking University, Beijing,
100871, China
[5]School of Geography, Earth and Environmental Sciences, University of Birmingham,
Birmingham, B15 2TT, UK
[*]Corresponding Email: liweijun@zju.edu.cn (W. J. Li)





**Abstract**
As one of the intense anthropogenic emission regions across the relatively high
latitude (> 40°N) areas on the Earth, Northeast China faces serious problem on regional
haze during long winter with half a year. Aerosols in polluted haze in Northeast China
are poorly understood compared with the haze in other regions of China such as North
China Plain. Here, we for the first time integrated bulk chemical measurements with
single particle analysis from transmission electron microscopy (TEM), nanoscale
secondary ion mass spectrometer (NanoSIMS), and atomic force microscopy (AFM)
to obtain morphology, size, composition, aging process, and sources of aerosol particles
collected during two contrasting regional haze events (Haze-I and Haze-II) at an
urban site and a mountain site in Northeast China, and further investigated the causes
of regional haze formation. Haze-I evolved from moderate (average $PM_{2.5}$: 76-108
$\mu g/m^3$) to heavy pollution (151-154 $\mu g/m^3$), with the dominant $PM_{2.5}$ component
changing from organic matter (OM) (39-45 $\mu g/m^3$) to secondary inorganic ions
(94-101 $\mu g/m^3$). Similarly, TEM observations showed that S-OM particles elevated
from 29% to 60% by number at urban site and 64% to 74% at mountain site and
75-96% of Haze-I particles included primary OM. Change of wind direction induced
that Haze-I rapidly turned into Haze-II (185-223 $\mu g/m^3$) with the predominant OM
(98-133 $\mu g/m^3$) and unexpectedly high $K^+$ (3.8 $\mu g/m^3$). TEM also showed that K-OM
particles increased from 4-5% by number to 50-52%. Our study revealed a contrasting
formation mechanism of these two haze events: Haze-I was induced by accumulation
of primary OM emitted from residential coal burning and further deteriorated by
secondary aerosol formation via heterogeneous reactions; Haze-II was caused by
long-range transport of agricultural biomass burning emissions. Moreover, we found
that 75-97% of haze particles contained tarballs, but only 4-23% contained black
carbon and its concentrations were low at 2.7-4.3 $\mu g/m^3$. The results highlight that
abundant tarballs are important light-absorbing brown carbon in Northeast China
during winter haze and further considered in climate models.



## 1. Introduction


Haze pollution is mainly caused by high levels of fine particulate matter ($PM_{2.5}$)
and it has widely spread over the globe such as Mexico city (Adachi and Buseck,
2008), Paris, France (Fortems‐Cheiney et al., 2016), North India (Chowdhury et al.,
2019), North China (Huang et al., 2014), and the Arctic (Frossard et al., 2011). In the
past twenty years, regional haze episodes with high concentrations of $PM_{2.5}$ have
frequently occurred in China following rapid economic development. Various studies
on regional haze in China have been conducted by many scientists (e.g., Bennartz et
al., 2011; Guo et al., 2014; Li and Shao, 2009; Lin et al., 2017; Liu et al., 2017b; Ren
et al., 2016; Shi et al., 2017; Zhang et al., 2018b). Current hot issues on haze pollution
concentrate on the formation processes of regional haze in various atmospheric
environments and their potential optical and health influences (Chen et al., 2017a;
Gao et al., 2017).
Many of these studies reported that haze particles can adversely affect human
health, ecological environments, and regional climate (Ding et al., 2016; Huang et al.,
2013; Lelieveld et al., 2015; Li et al., 2017a; Liu et al., 2019; Liu et al., 2016;
Mahowald et al., 2018; Shi et al., 2019; Xie et al., 2019; Zhang et al., 2013). For
example, exposure to high levels of ambient $PM_{2.5}$ cause or contribute to a variety of
human diseases (e.g., stroke, ischemic heart disease, chronic obstructive pulmonary
disease, and lung cancer) (Chen et al., 2019; Liu et al., 2016) and lead to ~1.3 million
premature deaths per year in China (Lelieveld et al., 2015). Abundant anthropogenic
metal, ammonium, and phosphorus-containing particles are transported into remote
regions of oceans, where they promote plankton growth and further influence ocean
ecology (Li et al., 2017a; Mahowald et al., 2018; Shi et al., 2019). High
concentrations of fine aerosols suspended in haze layers not only influence regional
climate through absorbing (e.g., black carbon (BC) and brown carbon (BrC)) and
scattering (e.g., sulfates and nitrates) solar radiation (Huang et al., 2013; Liu et al.,
2019; Xie et al., 2019), but also depress the planetary boundary layer development





(Ding et al., 2016; Li et al., 2017c; Zhang et al., 2013) and reduce crop yields (Tie et
al., 2016) in China. Therefore, understanding haze formation mechanisms and sources
of haze particles in different regions of China is crucial to provide feasible control
strategies for reducing regional $PM_{2.5}$ concentration and protecting ecosystems.

In China, regional heavy hazes frequently occur in the North China Plain (NCP)

(Zhao et al., 2013) and Northeast China (Ma et al., 2018) during winter. In the past
decades, the regional haze formation mechanisms in the NCP have been intensely
investigated (Chen et al., 2017b; Cheng et al., 2016; Li et al., 2015; Liu et al., 2017c;
Tao et al., 2014; Tian et al., 2015; Wang et al., 2016; Wang et al., 2019; Wang et al.,
2014; Xing et al., 2019; Zheng et al., 2015b; Zhong et al., 2019). Adverse
meteorological conditions (e.g., low wind speeds and stable atmospheric boundary
layer ) can induce preliminary formation of hazes during winter (Zheng et al., 2015b;
Zhong et al., 2019). Massive numbers of primary particles from industries,
households, and vehicular exhaust emissions (e.g., fly ash, metal, primary organic
matter, and soot particles) are the major aerosols in winter hazes (Chen et al., 2017b;
Tian et al., 2015; Wang et al., 2019). Rapid production of secondary aerosols (e.g.,
sulfates, nitrates, and secondary organics) via heterogeneous reactions under high RH
mainly elevates haze levels and causes regional hazes (Liu et al., 2017c; Wang et al.,
2016; Xing et al., 2019). The two-way feedback between accumulation of air
pollutants and depression of the atmospheric boundary layer also aggravate haze
pollution (Wang et al., 2014; Zhong et al., 2019). Moreover, regional transport of air
pollutants is one of the important factors for the long duration of regional haze (Li et
al., 2015; Tao et al., 2014; Zheng et al., 2015b). These studies well revealed formation
mechanisms of regional winter hazes in the NCP.

Among different regions or cities in China, the haze formation mechanisms

would most likely differ due to different emissions and meteorological conditions (Li
et al., 2019; Zhang et al., 2017; Zheng et al., 2015b). The air quality in Northeast
China, a region with a long heating period (mid October-mid April), is mainly



influenced by abundant inefficient combustion activities (e.g., coal and biomass burning in residential stoves and coal burning in small boilers for household heating and cooking) (Yang et al., 2017; Zhang et al., 2017). Extremely high concentrations of organic aerosols have been observed in Northeast China during winter haze that coincide with crop growing and harvest periods (Cao et al., 2017; Chen et al., 2015; Zhang et al., 2017). Because of the influences of the regional haze in winter, annual aerosol optical depth in urban areas was ~3.7 times higher than that in rural areas in Northeast China (Zhao et al., 2018b). In the past five years, some studies have focused on the physicochemical properties of haze particles collected in Northeast China (Cao et al., 2017; Chen et al., 2015; Li et al., 2017b; Miao et al., 2018; Zhang et al., 2017; Zhao et al., 2018b). However, studies on regional haze evolution and haze formation mechanisms in Northeast China are rare. This limited information precludes the comparison of regional hazes in Northeast China with other regions in China. Furthermore, it is difficult to adopt some reasonable regional haze pollution control strategies from the NCP to apply to the air pollution in Northeast China.

Northeast China is only one intense anthropogenic emission region besides Mongolia across the relatively high latitude (> 40°N) areas on the Earth (van Donkelaar et al., 2016). This region is significantly influenced by the Siberian cold high pressure systems in winter. The Siberian anticyclone transports air pollutants from Northeast China to Korea, Japan, and even the Arctic, and further causes large-scale influences in the global climate (Jung et al., 2015; Rodo et al., 2014; Sobhani et al., 2018; Zhang et al., 2016). Therefore, understanding the physicochemical characteristics of anthropogenic fine particles in regional haze and regional haze formation mechanisms over Northeast China has to be considered of the utmost importance.

In this study, we conducted a field experiment in the south part of Northeast China from 31 October to 6 November 2016. Two contrasting regional heavy haze episodes occurred there during the sampling period. We investigated the types and





mixing states of individual aerosol particles and PM$_{2.5}$ composition during these two
episodes. Finally, we elucidated the formation mechanisms of two regional haze
events and the main sources of fine haze particles.

**2. Experimental Methods**
**2.1 Sampling sites and sample collections**

There are three provincial capital cities (i.e., Shenyang, Changchun, and Harbin)

in Northeast China that are surrounded by the Greater Khingan Mountains, the Lesser
Khingan Mountains, and the Changbai Mountains (Figure 1a). In this study, we
selected an urban site (41.8°N, 123.35°E) and a mountain site (41.92°N, 123.65°E) in
Shenyang city (Figure 1b). The urban site is located in the center of Shenyang city. The
mountain site on top of Qipan Mountain (224 m) is located ~30 km northeast of the
urban site (Figure 1b). There are only a few villages around Qipan Mountain, so its air
quality well represents the regional transport in the south part of Northeast China.

The PM$_{2.5}$ samples were collected on 90 mm quartz filters (Whatman, UK) using

two medium volume samplers (Wuhan Tianhong Inc., TH-150A, 100 L/min) at the
urban and mountain sites. Individual particle samples were collected on transmission
electron microscopy (TEM) grids and silicon wafers using two DKL-2 samplers
(Genstar Inc., 1 L/min) equipped with a 0.5 mm jet nozzle impactor at two sampling
sites. The quartz filters provided mass concentrations and the chemical composition of
ambient PM$_{2.5}$. The TEM grids and silicon wafers were used for microscopic
observations of individual particles. To better explore the variation of PM$_{2.5}$
composition and individual particles, we collected the daytime (DT, 8:30-20:00 (local
time)) PM$_{2.5}$ and nighttime (NT, 20:30-8:00 (next day)) PM$_{2.5}$ as well as individual
particle samples four times a day (i.e., 0:00-3:00, 6:00-9:00, 12:00-15:00, and
18:00-21:00). To avoid individual particles overlapping on the substrate, the sampling
duration of individual particles was varied from 30 s to 10 min depending on the PM$_{2.5}$
concentrations. After individual particles samples were collected, we immediately





used a portable optical microscope to check the particle distribution on the substrate,
which guaranteed the sample to be suitable for microscopic analyses. After the
sampling, the quartz filters were put into a -20 °C refrigerator and the TEM grids and
silicon wafers were stored in the dry, clean, and airtight containers until laboratory
analyses were performed.
Meteorological data (i.e., wind speed and wind direction, temperature, relative
humidity (RH), and pressure) at the urban and mountain sites were simultaneously
collected by two automated weather instruments (Kestrel 5500, USA) at five minute
intervals.

**2.2 PM$_{2.5}$ analyses**
The quartz filters are weighed with a high-precision digital balance (Sartorius ME
5-F, 0.001 mg of reading precision) after being equilibrated for 24 h under stable
conditions (TP: 20±1 °C; RH: 48±2%) before and after sampling. The PM$_{2.5}$ mass
concentrations at the urban and mountain sites are calculated based on the weight
difference and sampling volume of each quartz filter.
Each quartz filter collected at two sampling sites can be used to analyze chemical
composition (i.e., water-soluble ions, organic carbon (OC), and elemental carbon
(EC)) of PM$_{2.5}$. In this study, we use an ion chromatography system (Dionex ICs-90,
USA) to obtain mass concentrations of water-soluble ions (i.e., Ca$^{2+}$, Mg$^{2+}$, K$^+$, Na$^+$,
NH$_4^+$, NO$_3^-$, SO$_4^{2-}$, Cl$^-$, and F$^-$) and an OC/EC analyzer (Sunset Laboratory Inc., USA)
to obtain mass concentrations of OC and EC. Concentrations of organic matter (OM)
were further calculated through multiplying OC concentrations by a factor of 1.4
reported by Guinot et al. (2007). The experimental details about water-soluble ions
analysis and OC/EC analysis have been provided in our previous study (Zhang et al.,

2017).


**2.3 TEM/EDS analysis**



Individual particles on TEM grids (copper (Cu) grid covered by a carbon (C)
reinforcement substrate) are examined using TEM combined with energy-dispersive
X-ray spectrometry (EDS) (JEOL, JEM-2100). TEM observation provide the
morphology of individual particles and the mixing states of different aerosol
components in individual particles on the substrate. EDS determines the elemental
composition of individual particles. EDS spectra of individual particles are collected
within a maximum time of 30 s to minimize potential X-ray damage and ensure
sufficient intensity during EDS analysis. Cu cannot be analyzed for individual particles
because the TEM grids are made of Cu. In addition, the C content in EDS spectra of
individual particles may be overestimated due to the substrate's contribution.
Individual particles are unevenly distributed on TEM grids, with coarser particles
in the center of sampling spot and finer particles on the periphery. Therefore, to
guarantee that the analyzed particles are representative, five areas are selected from the
sampling center to periphery on each TEM grid. After a labor-intensive operation, a
total of 3,630 particles at the urban site and 4,281 particles at the mountain site with
diameter < 2.5 μm were analyzed by TEM/EDS.
The area, perimeter, and equivalent circle diameter (ECD) of individual particles
in TEM images are manually or automatically obtained through an image analysis
software (iTEM, Olympus soft imaging solutions GmbH, Germany).

**2.4 NanoSIMS analysis**
Based on TEM/EDS analysis, a representative individual particle sample is
selected for nanoscale secondary ion mass spectrometer (NanoSIMS, 50L, CAMECA
Instruments, Geneviers, France) analysis. In this work, signal intensity mapping of
$^{12}C^{14}N^-$, $^{12}C^-$, $^{16}O^-$, and $^{32}S^-$ ions are obtained after the $Cs^+$ primary ion beam ionizes
the atoms of particle surface. Ion signal intensity mapping of individual particles with
nanometer resolution clearly shows ion distribution in particles. Strong $^{12}C^{14}N^-$ and
$^{12}C^-$ signals imply OM in individual particles and eliminate the contribution of C



substrate on TEM grids (Chi et al., 2015).

**2.5 AFM analysis**
The surface structure of individual particles collected on silicon wafers are
investigated using atomic force microscopy (AFM, Dimension Icon, USA) with a
digital nanoscope IIIa in the tapping mode. AFM observations in the tapping mode
produce 3-D images of individual particles. The tapping AFM is equipped with a
cantilever and a conical tip with a 10 nm radius.
Based on the preliminary observations from TEM, we select three typical samples
collected during the clean day, Haze-I event, and Haze-II event for the AFM analysis
(The naming details for haze events in Section 3.1). To obtain the 3-D morphology of
more particles on the basis of keeping images clear, $10 \times 10$ $\mu m^2$ of scanning range and
0.5-0.8 Hz of scanning rate are selected. 57 particles during the clean day and Haze-I
and 29 particles during the Haze-II were carefully analyzed. After obtaining AFM
images of individual particles, we use the NanoScope Analysis software to
automatically obtain the bearing area (A) and bearing volume (V) of each analyzed
particle. The ECD and equivalent sphere diameter (ESD) of individual particles are
further calculated according to the following two formulas.
$$A = \frac{4}{3}\pi r^2 = \frac{\pi d^2}{3} \rightarrow d = \sqrt{\frac{3A}{\pi}} \qquad (1)$$
$$V = \frac{4}{3}\pi r^3 = \frac{4}{3} \times \frac{\pi D^3}{8} \rightarrow D = \sqrt[3]{\frac{6V}{\pi}} \qquad (2)$$
Where $d$ is ECD; $D$ is ESD.
The linear correlations between ECD and ESD and typical AFM images of
individual particles (clean day & Haze-I: D=0.5861×d; Haze-II: D=0.4040×d) are
shown in Figures S1a-b. Through the above two linear correlations, we can obtain the
ESD of each individual particle analyzed by iTEM software.





## 3. Results

### 3.1 Composition of fine particles and meteorology

Regional haze pollution was observed in Northeast China during the sampling period from 31 October to 5 November 2016 (Figure S2). Based on the variation of $PM_{2.5}$ mass concentration and visibility at the urban and mountain sites, clean day (31 DT October, $PM_{2.5} < 75$ μg/m$^3$ and visibility $> 10$ km) and haze days (31 NT October-5 NT November, $PM_{2.5} \geq 75$ μg/m$^3$ and visibility $< 10$ km) were identified (Figure 2a). In general, we determined two regional haze events: 31 NT October-4 DT November (Haze-I) and 4 NT-5 NT November (Haze-II), based on the different prevailing wind directions (Haze-I: Southerly; Haze-II: Northerly) (Figures S3a-b) and air mass backward trajectories (Figure S4).

To understand haze evolution, we divided Haze-I event into a moderate haze stage (31 NT October-3 NT November) with 75 μg/m$^3$ $\leq PM_{2.5} < 150$ μg/m$^3$ and a heavy haze stage (4 DT November) with $PM_{2.5} \geq 150$ μg/m$^3$ (Figure 2a). Figure 2a shows that the average mass concentrations of $PM_{2.5}$ increased from 69 μg/m$^3$ to 108 μg/m$^3$ at the urban site and from 25 μg/m$^3$ to 76 μg/m$^3$ at the mountain site from the clean day to the moderate Haze-I event. Compared with northerly winds with ~1.4 m/s and ~38% of RH on the clean day, the wind directions were from the southerly with 0.7 m/s at the urban site and 3.0 m/s at the mountain site; RH varied from 39% to 67% during the moderate Haze-I event (Figures S3a-d). Here the moderate Haze-I event is considered as a general haze pollution that frequently occurs in Northeast China during winter (Zhang et al., 2017). During the moderate Haze-I event, the average mass concentrations of OM, EC, and secondary inorganic ions (i.e., $SO_4^{2-}$, $NO_3^-$, and $NH_4^+$) in $PM_{2.5}$ were 45 μg/m$^3$, 4.3 μg/m$^3$, and 24 μg/m$^3$ at the urban site and 39 μg/m$^3$, 2.7 μg/m$^3$, and 20 μg/m$^3$ at the mountain site, respectively (Figures 2b-c). Following $PM_{2.5}$ concentration exceeding 150 μg/m$^3$, the moderate Haze-I event evolved into the heavy Haze-I event (Figure 2a). During the heavy Haze-I event, mass concentrations of $PM_{2.5}$ were 154 μg/m$^3$ at the urban site and 151 μg/m$^3$ at the



mountain site (Figure 2a). RH remained high at 73-80% at the urban and mountain sites
during the heavy Haze-I event (Figures S3c-d). Figures 2b-c show that the average
concentrations of OM and EC were fairly constant at 39-45 $\mu g/m^3$ and 2.7-4.3 $\mu g/m^3$
from the moderate Haze-I to the heavy Haze-I event. In contrast, secondary inorganic
ions rapidly increased from 20-24 $\mu g/m^3$ to 94-101 $\mu g/m^3$ in the DT of 4 November
(Figures 2b-c). Therefore, the variation of chemical composition of $PM_{2.5}$ clearly
reflected the general haze evolution from moderate to heavy in Northeast China.

With the prevailing wind changing from southerly with ~0.8 m/s to northerly

with ~3.9 m/s (Figures S3a-b), the Haze-I event turned into the Haze-II event. The
average $PM_{2.5}$ concentrations remained at high levels and reached 223 $\mu g/m^3$ at the
urban site and 185 $\mu g/m^3$ at the mountain site (Figure 2a). RH were consistently high in
the range of 65-87% during the Haze-II event (Figures S3c-d). In addition, we noticed
that hourly concentrations of $PM_{2.5}$ and CO rapidly climbed from 209 $\mu g/m^3$ and 1.3
ppm at 6:00 A.M. to 669 $\mu g/m^3$ and 1.9 ppm at 8:00 A.M. on 5 November, respectively
(Figures S5a-b). Although $PM_{2.5}$ concentrations during the Haze-II event were close to
those of the heavy Haze-I event, we found large differences between aerosol
chemistry in these two heavy haze episodes besides the prevailing wind direction
(Figures 2a-c). From the heavy Haze-I to the Haze-II events, secondary inorganic ions
significantly decreased from 62-66% of the total $PM_{2.5}$ mass (94-101 $\mu g/m^3$) to 31-35%
(65-70 $\mu g/m^3$); but OM markedly increased from 27-30% (42-45 $\mu g/m^3$) to 53-60%
(98-133 $\mu g/m^3$) (Figures 2a-c). In addition, $K^+$ average concentrations unexpectedly
increased from 1.4 $\mu g/m^3$ during the heavy Haze-I event to 3.8 $\mu g/m^3$ during the
Haze-II event (Figures 2b-c). As a result, the difference between $PM_{2.5}$ composition
before and after 4 November again proved that there were two different haze episodes
under the different prevailing wind directions. These two regional haze episodes
might have different formation mechanisms (details in Section 4.1).

**3.2 Characteristics of individual haze particles**



TEM/EDS did an excellent job of determining the morphology and composition
of individual particles as shown in Figures 3a-j. NanoSIMS was used to further
identify OM particles through ions ($^{12}C^{14}N^-$, $^{12}C^-$, $^{16}O^-$, and $^{32}S^-$) signal mappings in
order to exclude the interfere of C substrate on TEM grids to EDS (Figure 3k). Figure
3k shows high $^{12}C^{14}N^-$ and $^{12}C^-$ signals but low $^{32}S^-$ signal, which strongly confirms
the OM particle. As a result, six basic types of aerosol components were classified
based on their morphology, composition, and ion signal mapping: mineral, OM, soot
(also known as EC and BC), fly ash/metal, S-rich, and K-rich (Figures 3a-k).
Mineral particles mainly contain O, Si, Al, and Fe elements and present irregular
shape (Figures 3a-1 and 3a-2). Mineral particles mainly occurred in the coarse size
range (> 1 μm) (Figures S6a-b), and they were often externally mixed with other
types of particles (i.e., OM, soot, fly ash/metal, S-rich, and K-rich particles) (Figure
3a-1). OM particles are mainly composed of C, O, and Si (Figures 3b-2, 3c-2, and
3i-2). OM particles were further classified into spherical OM (Figure 3b-1), domelike
OM (Figure 3c-1), and OM coating (Figures 3i-1 and 3j) based on their morphology.
TEM observations showed that most OM particles were internally mixed with S-rich
(Figure 3f-1), soot (Figure 3g), and K-rich (Figures 3i-1 and 3j) particles. Soot
particles mainly include C and O elements and are aggregates of carbonaceous
spheres (Figures 3d-1 and 3d-2). Figures S6a-b show that soot particles mainly
occurred in the fine size range (< 200 nm). TEM observations showed that a majority
of soot particles were internally mixed with S-rich or OM particles, which were
classified as soot-S/OM particles (Figure 3g). Fly ash/metal particles with spherical
morphology are mainly comprised of O, Si, and metallic elements (e.g., Al, Fe, Mn,
and Pb) (Figures 3e-1 and 3e-2). Fly ash/metal particles were mainly in the ultrafine
size range (< 100 nm) (Figures S6a-b) and internally mixed with S-rich or OM
particles, which were called fly ash/metal-S/OM particles (Figure 3h). S-rich particles
are mainly composed of O, S, and N (Figure 3f-2) and formed from the oxidation of
$SO_2$, $NO_x$, and $NH_3$. S-rich particles normally represent the mixtures of $(NH_4)_2SO_4$



and $NH_4NO_3$ (Li et al., 2016). TEM observations showed that abundant S-rich
particles were internally mixed with OM particles, called S-OM particles (Figure
3f-1). K-rich particles mainly contain K, O, S, and N elements (Figure 3i-3). All
K-rich particles were internally mixed with OM particles, and were called K-OM
particles (Figures 3i-1 and 3j). To compare number fractions of OM particles during
the haze evolution, here we considered the particles including OM as OM-containing
particles. In the same way, soot-containing and fly ash/metal-containing particles were
also defined.

Figure 4 shows the variation of number fractions of different particles types at

the urban and mountain sites from the clean day to the Haze-I event and to the Haze-II
event. At the urban site, these data show that mineral and S-OM were the major
particle types, which accounted for 36% and 23% during the clean day. Subsequently,
S-OM, OM, and soot-containing particles became major particle types which
accounted for 29%, 18%, and 23% at the urban site during the moderate Haze-I event.
During the heavy Haze-I event, S-OM particles dominated at 60% at the urban site
(Figure 4). Interestingly, we found that S-OM particles remained at very high
frequencies (61-74%) at the mountain site from the clean day to the Haze-I event
(Figure 4). Furthermore, the pink frames in Figure 4 show that OM-containing (i.e.,
S-OM, OM, soot-OM, fly ash/metal-OM, and K-OM) particles accounted for 75-86%
at the urban site and 95-96% at the mountain site during the Haze-I event. During the
Haze-II event, number fractions of OM-containing particles reached their maximum
at 96% at the urban site and 97% at the mountain site (Figure 4). It is noted that
K-OM became the predominant particles in the Haze-II event, accounting for 50% at
the urban site and 52% at the mountain site (Figure 4). Therefore, individual particle
analysis clearly shows large differences of particle types and fractions following the
haze evolution, which is consistent with the variation of $PM_{2.5}$ composition described
in Section 3.1.



### 3.3 Pollutants change following haze evolution and transformation


The analysis of X/EC (e.g., $PM_{2.5}/EC$, OC/EC, $SO_4^{2-}/EC$, $NO_3^-/EC$, and $K^+/EC$)
factors not only can exclude the influence of changes in atmospheric boundary layer
height on pollutants mass concentrations, but also indicate accumulation (e.g., small
changes in X/EC) and secondary formation (e.g., increases of $SO_4^{2-}/EC$ and $NO_3^-/EC$)
of $PM_{2.5}$ during haze evolution (Zhang et al., 2017; Zheng et al., 2015b). Although
$PM_{2.5}$ concentrations increased from 69 $\mu g/m^3$ to 108 $\mu g/m^3$ at the urban site and from
25 $\mu g/m^3$ to 76 $\mu g/m^3$ at the mountain site from the clean day to the moderate Haze-I
event, X/EC factors only displayed minor variations (Figures 5a-b). This result shows
that accumulation of air pollutants mainly induced the moderate Haze-I formation.
Following formation of the heavy Haze-I event with $PM_{2.5}$ concentrations at 154
$\mu g/m^3$ at the urban site and 151 $\mu g/m^3$ at the mountain site, $SO_4^{2-}/EC$, $NO_3^-/EC$, and
$PM_{2.5}/EC$ factors dramatically increased (Figures 5a-b). Figures 5a-b show that
$SO_4^{2-}/EC$ and $NO_3^-/EC$ factors reached their maximum values at the two sampling
sites during the heavy Haze-I event, suggesting that massive secondary sulfates and
nitrates formed during Haze-I evolution. In contrast, $SO_4^{2-}/EC$ and $NO_3^-/EC$ factors
began to decrease, but OC/EC, $K^+/EC$, and $PM_{2.5}/EC$ factors significantly increased
during the Haze-II event, although $PM_{2.5}$ concentrations became even further elevated
to 223 $\mu g/m^3$ at the urban site and 185 $\mu g/m^3$ at the mountain site (Figures 5a-b). This
result indicates that large amounts of aerosol including OM and $K^+$ contributed to the
conversion from the Haze-I to the Haze-II events.
Individual particle analysis shows consistent results with the variation of X/EC
factors described above: the fractions of OM, S-OM, and soot-containing particles
only had minor changes at 6-9% (i.e., 9% to 18%, 23% to 29%, and 14% to 23%) at
the urban site and at 2-3% (i.e., 9% to 11%, 61% to 64%, and 21% to 18%) at the
mountain site from the clean day to the moderate Haze-I event (Figure 4). However,
along with the moderate Haze-I evolving into the heavy Haze-I event, S-OM fractions
suddenly increased from 29% to 60% at the urban site and from 64% to 74% at the



mountain site (Figure 4). When the heavy Haze-I turned into the Haze-II events,
S-OM fractions significantly decreased from 60-74% to 30-32% at two sampling sites
but K-OM fractions largely increased from 4-5% to 50-52% (Figure 4). As a result,
the chemical data of PM$_{2.5}$ samples and individual particle analysis both well reflect
the haze evolution and transformation.

**4. Discussion**
**4.1 Sources and formation of two distinctive haze events**

Our analyses show that the fractions of OM in PM$_{2.5}$ by mass (35-41%, Figures

2a-c) and OM-containing in individual particles by number (> 70%, Figure 4) both
were elevated during the Haze-I event at the urban and mountain sites. TEM
observations showed two major types of OM particles: spherical OM and domelike
OM during the Haze-I event (Figures 6c-d). These OM particles have been considered
as primary OM aerosol, which can indicate the direct emissions from residential coal
burning (Zhang et al., 2018a) or biomass burning (Liu et al., 2017a). In addition, we
noticed that K-OM particles as a tracer of biomass burning (Adachi and Buseck, 2008;
Bi et al., 2011; Liu et al., 2017a) were quite low at ~3% by number fraction during the
Haze-I event (Figure 4) and had poor correlations with PM$_{2.5}$ concentrations (Figure
7a). Therefore, we can exclude biomass burning and conclude that coal burning was
the major contribution to emissions during the Haze-I event. During the wintertime
with the lowest temperature at about -30 °C in Northeast China, high-intensity coal
burning activities for household heating are necessary (Xu et al., 2017; Zhang et al.,
2017). Central heating through large boilers equipped with efficient filters are in wide
use in the urban areas in Northeast China, but residential stoves without emission
controls are still used for household heating and cooking in the rural and suburban
areas (Zhang et al., 2017). Through the meteorological data (Figures S3a-b) and air
mass backward trajectory (Figure 7c) analyses, we inferred that the air at the two
sampling sites during the Haze-I event was mostly influenced by Shenyang city and its



nearby surrounding emissions. As a result, we determined that large amounts of fine
primary OM aerosol during the Haze-I event were mainly from regional emissions of
coal burning in residential stoves for heating and cooking as shown in Figure 8.

The significant increases of $SO_4^{2-}$/EC, $NO_3^-$/EC, and $PM_{2.5}$/EC factors from the

moderate to heavy Haze-I event (Figures 5a-b) suggest that massive secondary
production of sulfates and nitrates elevated $PM_{2.5}$ concentrations. Similar results that
secondary aerosol formation causes heavy winter hazes in the NCP have been well
documented (Liu et al., 2017c; Sun et al., 2014; Xue et al., 2016; Zheng et al., 2015a).
Although photochemical activity is weak because of thick haze layers weakening solar
radiation (Zhao et al., 2013; Zheng et al., 2015b), heterogeneous chemical reactions on
particle surfaces have been considered as major pathways in the formation of sulfates
and nitrates from $SO_2$ and $NO_x$ whenever RH exceeds 70% (Sun et al., 2018; Wang et
al., 2016; Wu et al., 2018). Here, TEM observations did show that large numbers of
primary OM particles were coated by S-rich aerosols at 73-80% of RH during the
heavy Haze-I event (Figures 6c-d). Obviously, the preexisting particles in the hazes
provided large heterogeneous surfaces for the formation of sulfates and nitrates (He et
al., 2014; Li et al., 2011; Zhang et al., 2017). When the moderate Haze-I turned into
the heavy Haze-I event, we indeed found that secondary inorganic ions (i.e., $SO_4^{2-}$,
$NO_3^-$, and $NH_4^+$) became the predominant species in $PM_{2.5}$, accounting for 62-66%
(Figures 2a-c). Therefore, we conclude that atmospheric heterogeneous reactions
under high RH (> 70%) are a critical factor in heavy haze formation in Northeast
China.

In contrast to the Haze-I event, we found that concentrations of $K^+$ and OM in

$PM_{2.5}$ both were elevated about twice as much during the Haze-II event (Figures 2b-c).
In addition, TEM observations showed that the fraction of K-OM particles
explosively increased by 45-48% during the Haze-II event (Figure 4). High levels of
$K^+$ or K-OM particles represent the influences of biomass burning (Bi et al., 2011; Liu
et al., 2017a). Indeed, MODIS data during the Haze-II event show that many fire





spots occurred in the north part of Northeast China, far away ~580 km from sampling
sites, and the air mass backward trajectories at the urban and mountain sites both
crossed these fire spots (Figure 7d). MODIS maps further show that these fire spots
were in the farming lands instead of the forest areas (Figure 7d). Based on the MODIS
data and reports from the local department of ecology and environment
(http://www.hljdep.gov.cn/hjgl/zfjc/jphz/2017/05/15698.html), these dense fire spots
on 4 November were agricultural biomass burning. Open agricultural biomass burning
is normally intense during the early winter in Northeast China, a region with abundant
agricultural production, because large amounts of agricultural waste need to be
cleaned up and their burnt products can be used as fertilizers to increase soil fertility
(Cao et al., 2017; Yang et al., 2017). Along with the backward trajectories of air
masses, we found sharp $PM_{2.5}$ concentrations occurring from Harbin city (1281 μg/m$^3$
at 18:00 on 4 November) to Changchun city (658 μg/m$^3$ at 0:00 on 5 November) and
to Shenyang city (669 μg/m$^3$ at 8:00 on 5 November) (Figure S5a). These above
analyses well indicated that long-range transport of agricultural biomass burning
emissions directly led to the Haze-II formation (Figure 8) and explained why $K^+$/EC,
OC/EC, and $PM_{2.5}$/EC factors together reached their maximum values during the
Haze-II event (Figures 5a-b) and K-OM fractions and $PM_{2.5}$ mass concentrations had
strong positive correlation (Figure 7b). Overall, although the Haze-II and Haze-I
events both displayed heavy pollution levels, they had contrasting emission sources
and formation mechanisms as illustrated in Figure 8.

**4.2 Comparison: haze in Northeast China vs. NCP**

OC/EC ratios in $PM_{2.5}$ of 8.0-10.6 during the Haze-I event and 25.4-27.9 during
the Haze-II event were reported in Northeast China in this study (Table S1). The
results are much higher than OC/EC ratios in $PM_{2.5}$ during winter haze episodes in the
NCP, such as 4.5 in Beijing city (Zhao et al., 2013), 5.5 in Jinan city (Chen et al.,
2017b), and 5.3 in Tianjin city (Han et al., 2014). In addition, TEM observations





showed that OM-containing particles were dominant by number fraction (e.g., 75-96%
during the Haze-I event and 96-97% during the Haze-II event) in Northeast China
(Figure 4). The results are higher than 50-70% during regional haze episodes in the
NCP which were influenced by residential coal burning (Chen et al., 2017b; Li et al.,
2012). These comparisons suggest that the contribution of residential coal burning and
biomass burning to winter haze formation in Northeast China is significantly larger
than that in the NCP. Moreover, fly ash/metal-containing particles as the tracers of
industry emissions frequently occurred in regional hazes in the NCP, such as ~30% by
number fraction in Jinan city (Li et al., 2011), 23-29% in Beijing city (Ma et al., 2016),
and 13-17% in Xianghe (Zou et al., 2019). These data are much higher than 8%
(Haze-I) and 1% (Haze-II) reported in this study (Figure 4), suggesting that the
contribution of industrial emissions to regional haze in Northeast China is much
smaller than that in the NCP. Many studies have summarized formation mechanisms
of general regional haze in North China: regional transport of air pollutants (Zheng et
al., 2015b), massive formation of secondary aerosols (Shao et al., 2019), and two-way
feedback between adverse meteorological conditions and pollutant accumulation
(Zhong et al., 2019). We found that the Haze-I formation in this study can be attributed
to these three formation mechanisms (Figure 8). Moreover, the Haze-II formation
indicates that the primary emissions from intensive agricultural biomass burning
activities can rapidly induce regional heavy haze formation in Northeast China
(Figure 8).

**5. Atmospheric implications**
High concentrations of $PM_{2.5}$ in haze adversely affect human health by inducing
various respiratory diseases (Cohen et al., 2017; Liu et al., 2016). In China, long-term
exposure to high levels of ambient $PM_{2.5}$ has resulted in 1.1-1.35 million premature
deaths (Cohen et al., 2017; Lelieveld et al., 2015). The large-scale coal burning for
household heating and cooking in Northeast China emits vast numbers of



OM-containing particles (discussed in Section 4.1), which include abundant
polycyclic aromatic hydrocarbons (PAHs) (Chen et al., 2017b; Chen et al., 2013;
Zhang et al., 2008). Based on Ebenstein et al. (2017), people's life expectancy
decreases by 0.64 years for every 10 μg/m$^3$ contributed by residential coal burning
emissions. Furthermore, we found that 1-8% of haze particles contained toxic metals
(e.g., Pb, Zn, Cr, and Mn) from fly ash/metal particles with sizes smaller than 100 nm
in Northeast China (Figures 4 and S6a-b). Oberdorster et al. (2004) indicated that
such ultrafine metal particles can exert adverse health impacts via deposition in
human lungs and further penetration into the blood. Therefore, these OM-containing
and fly ash/metal-containing particles observed by TEM (Figures 6a-f) are harmful to
human health in Northeast China in winter.
The light-absorbing particles (i.e., BC and BrC) have been proved to reduce solar
radiation reaching the ground and further influence regional climate and crop
production (Alexander et al., 2008; Bond et al., 2013; Tie et al., 2016; Wang et al.,
2018). This study shows that concentrations of BC (i.e., EC or soot) were low at
2.7-4.3 μg/m$^3$ during the haze days and only accounted for 1.4-3.9% of PM$_{2.5}$ in
Northeast China (Figures 2a-c). These results are lower than them in other regions of
China such as the NCP: 7.6 μg/m$^3$ and 5.2% in Beijing city (Wu et al., 2016), Yangtze
River Delta: 11.8 μg/m$^3$ and 4.4% in Shanghai city (Zhi et al., 2014), and Pearl River
Delta: 9.6 μg/m$^3$ and 7.7% in Guangzhou city (Tao et al., 2012). These comparisons
indicate that BC concentrations are at low levels in Northeast China during winter
haze periods, although PM$_{2.5}$ concentrations are extremely high. Furthermore, TEM
observations showed that 75-97% of haze particles were OM-containing particles
such as spherical OM and domelike OM, but only 4-23% were soot-containing
particles in Northeast China (Figure 4). Previous studies have demonstrated that these
spherical and domelike OM particles from coal and biomass burning with low
combustion efficiency are tarballs (Chakrabarty et al., 2010; Hand et al., 2005; Pósfai
et al., 2004; Zhang et al., 2018a) or precursors of tarballs (Adachi et al., 2019;





Sedlacek III et al., 2018; Tóth et al., 2014), respectively. Tarballs as the primary BrC
have a wide absorption spectrum from visible to ultraviolet wavelengths (Hoffer et al.,
2016). The comparison in Section 4.2 indicates that the highest emission of tarballs in
China during winter occurs in Northeast China because of the intense and inefficient
residential coal burning and agricultural biomass burning. Therefore, how large
numbers of tarballs in regional winter haze influence atmospheric optical radiation
and climate in Northeast China should be studied in much greater detail.

Implementing suitable policies to encourage cleaner energy use (e.g., natural gas

and electricity) instead of coal and biomass for household heating are necessary in
Northeast China. Such policies would not only reduce regional haze formation but
also reduce the toxic aerosol components in the air. Admittedly, this transition of
heating energy would be facing a monumental challenge in Northeast China due to its
low economic growth and its low urbanization development rates (Zhao et al., 2018a).
Moreover, how to control open agricultural biomass burning should be considered in
Northeast China. Over the past decades, agricultural biomass returned to soils and
recycling agricultural biomass have both failed in Northeast China. This failure arises
because agricultural biomass does not decay throughout the winter with its low
temperatures down to -30 °C and snow or ice covers and its recycling consumes a lot
of manpower, material resources, and money (Yan et al., 2006). Therefore, how to
deal with agricultural biomass waste in Northeast China should be carefully thought
out by the local farmers in concert with the government.

**6. Conclusions**

To understand the formation mechanisms of winter haze in Northeast China, we

carried out an aerosol experiment at an urban site and a mountain site in the south part
of Northeast China from 31 October to 6 November 2016. Two different regional
heavy haze events (Haze-I and Haze-II) were identified during the sampling period.
Chemical composition (water-soluble ions, OM, and EC) of $PM_{2.5}$ were obtained using



ion chromatography and OC/EC analysis. Types and mixing states of individual
particles were identified using TEM/EDS and NanoSIMS: mineral, fly
ash/metal-containing, soot-containing, OM, S-OM, K-OM, and OM-containing.
Haze-I event has an evolution process from moderate to heavy pollution. OM was
the dominant component in $PM_{2.5}$ and their concentrations were 45 $\mu g/m^3$ at the urban
site and 39 $\mu g/m^3$ at the mountain site during the moderate Haze-I event. Individual
particle analysis also showed that over 70% of particles contained OM during the
moderate Haze-I event. Following the Haze-I evolution, secondary inorganic ions (i.e.,
$SO_4^{2-}$, $NO_3^-$, and $NH_4^+$) became the dominant components (94-101 $\mu g/m^3$) during the
heavy Haze-I event with $PM_{2.5}$ concentrations of 151-154 $\mu g/m^3$. Similarly, the number
fractions of S-OM increased from 29% to 60% at the urban site and from 64% to 74%
at the mountain site from the moderate Haze-I to the heavy Haze-I event. Along with
the prevailing wind suddenly changing from southerly with ~0.8 m/s to northerly with
~3.9 m/s, the heavy Haze-I rapidly turned into the Haze-II events with $PM_{2.5}$
concentrations of 185-223 $\mu g/m^3$. Meanwhile, OM replaced secondary inorganic ions
as the dominant component in $PM_{2.5}$ during the Haze-II event, accounting for 53-60%.
Furthermore, $K^+$ concentration during the Haze-II event was about three times higher
than that during the heavy Haze-I event. Individual particle analysis showed
consistent results that the number fractions of K-OM significantly increased from 4-5%
to 50-52%.
Based on our study, the accumulation of primary OM particles, mainly emitted
from residential coal burning, induced the moderate Haze-I formation with the onset of
stable meteorological conditions. Production of secondary aerosols via heterogeneous
reactions at high RH (> 70%), in particular sulfates and nitrates, caused the transition
from the moderate Haze-I to the heavy Haze-I. Furthermore, the long-range transport
of primary emissions from intense agricultural biomass burning led to the Haze-II
formation in the downwind areas in Northeast China. Our study also reveals that the
significant light-absorbing particles in Northeast China during winter are tarballs,





which is different from other regions in China influenced by serious air pollution.





**Author contributions**

**Author contributions**
JZ and WL conceived the study and wrote the manuscript. The field campaign was
organized and supervised by JZ and WL, and assisted by LL, LX, and HZ. JZ, LL, LX,
and QL contributed the sample analyses. All authors reviewed and commented on the
paper.

**Acknowledgements**
We appreciate Peter Hyde's comments and proofreading. We thank Jiaxing Sun
and Yong Ren for their assistance of sample collections. This work was funded by the
National Key R&D Program of China (2017YFC0212700), National Natural Science
Foundation of China (91844301, 41622504, 41575116), Zhejiang Provincial Natural
Science Foundation of China (LZ19D050001), and Zhejiang University Education
Foundation Global Partnership Fund (188170-11103/004). All data presented in this
paper are available upon request from the corresponding author (W. J. Li, e-mail:
liweijun@zju.edu.cn).



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

**Figure 8.** Schematic diagram of Haze-I and Haze-II formation in Northeast China
during winter. The major emission sources and haze formation processes are shown.





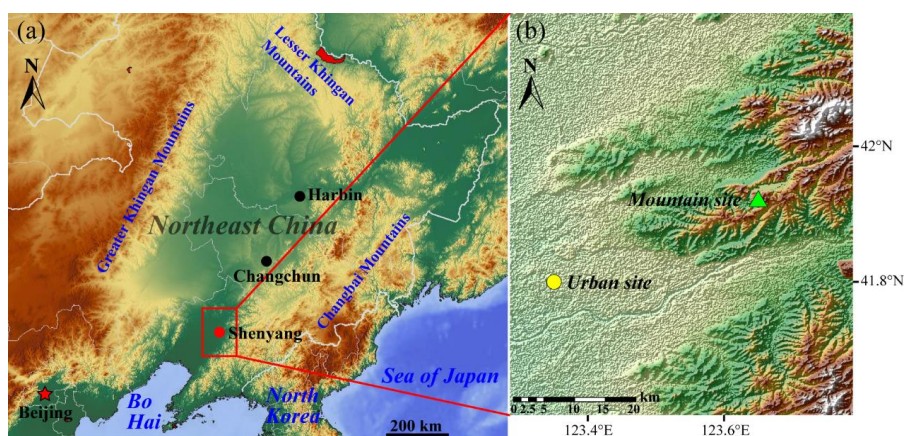


**Figure 1.**



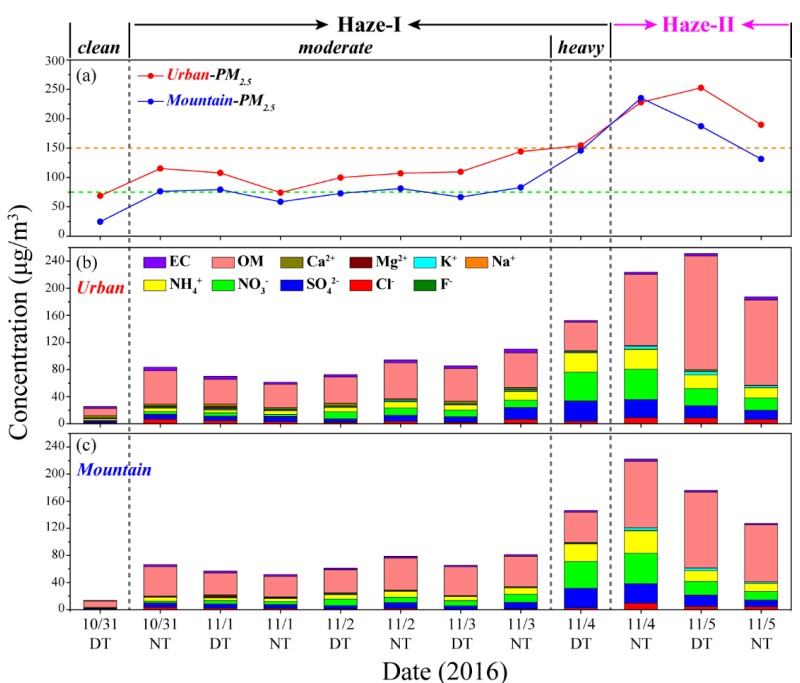


**Figure 2.**



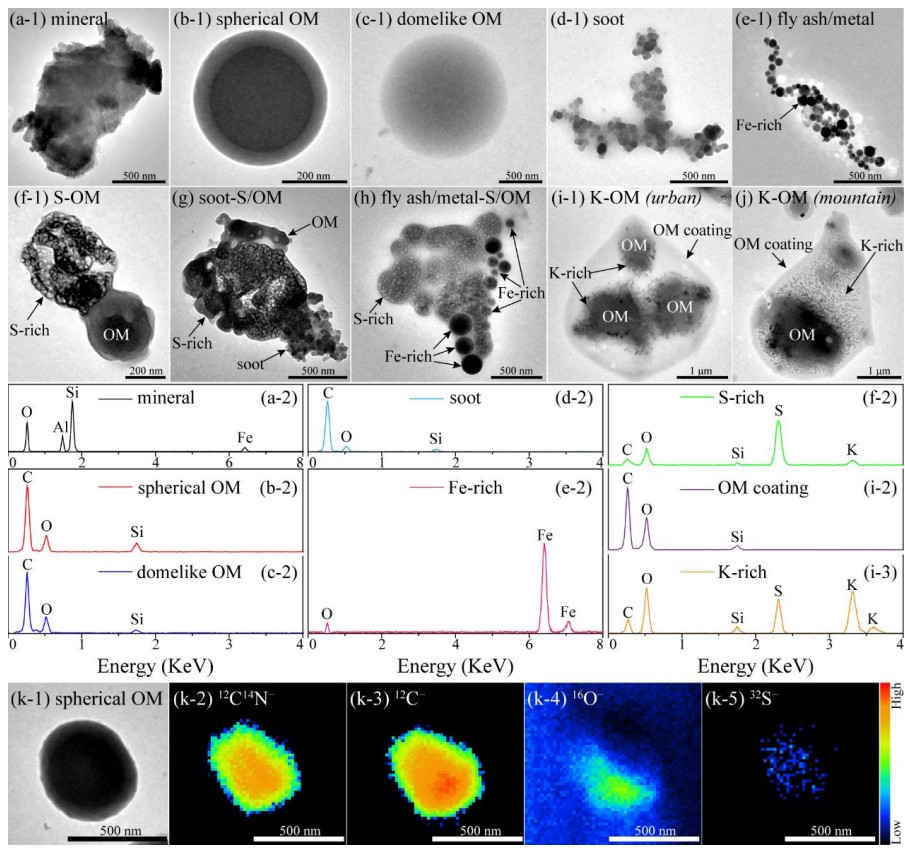

**Figure 3.**





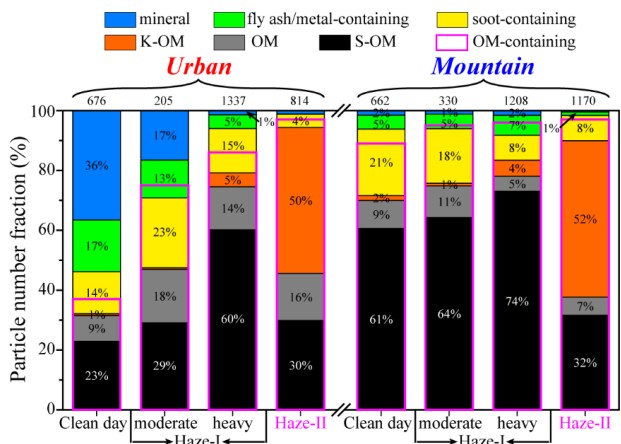


**Figure 4.**





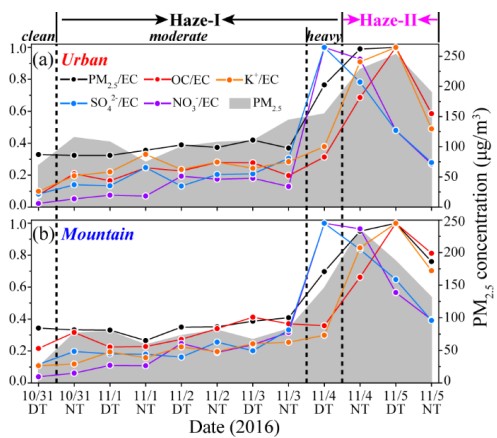


<p align="center">**Figure 5.**</p>

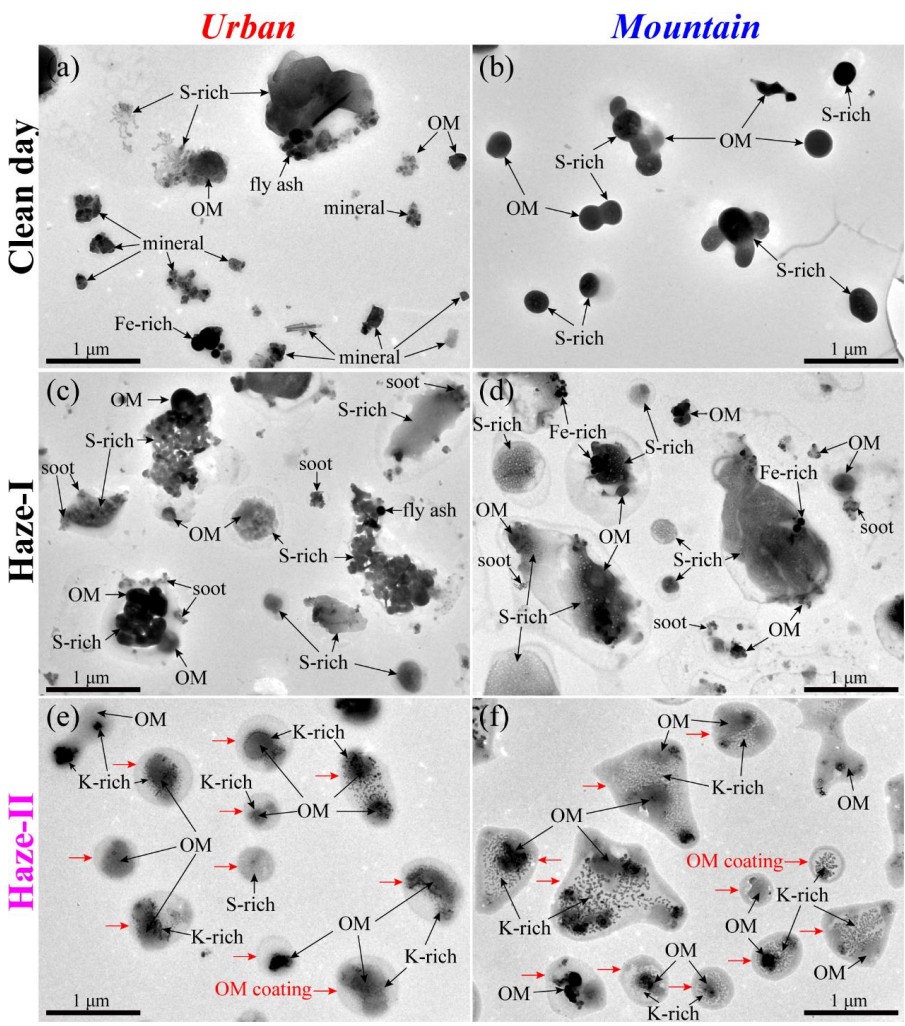


**Figure 6.**





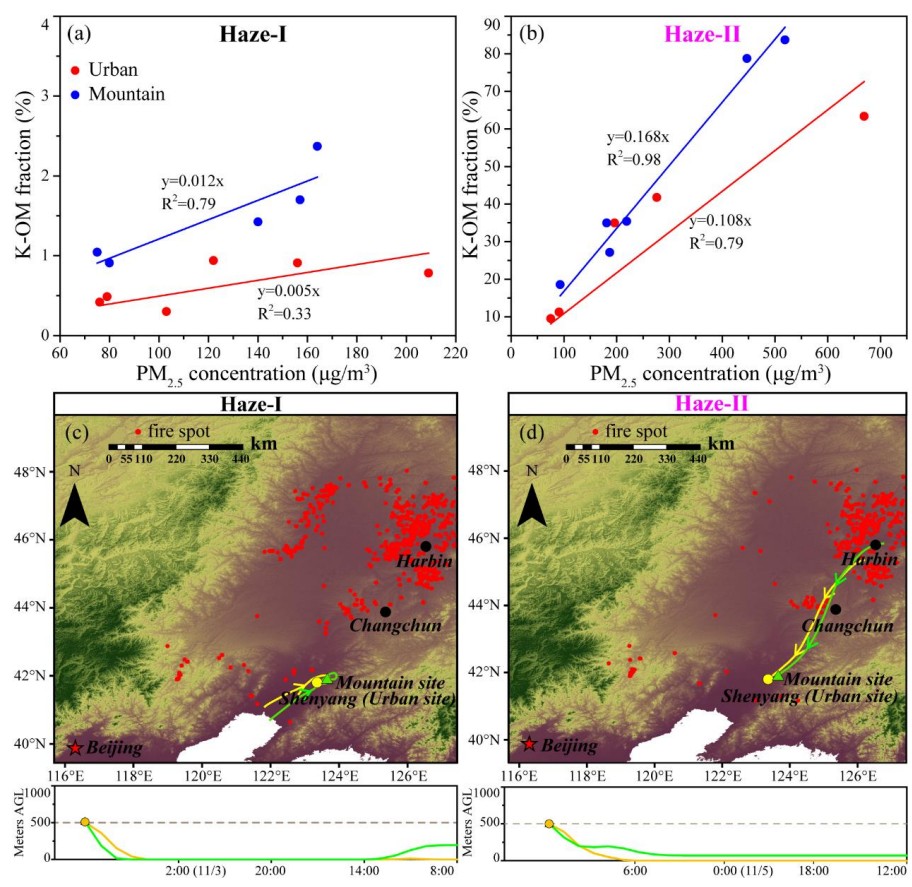


**Figure 7.**



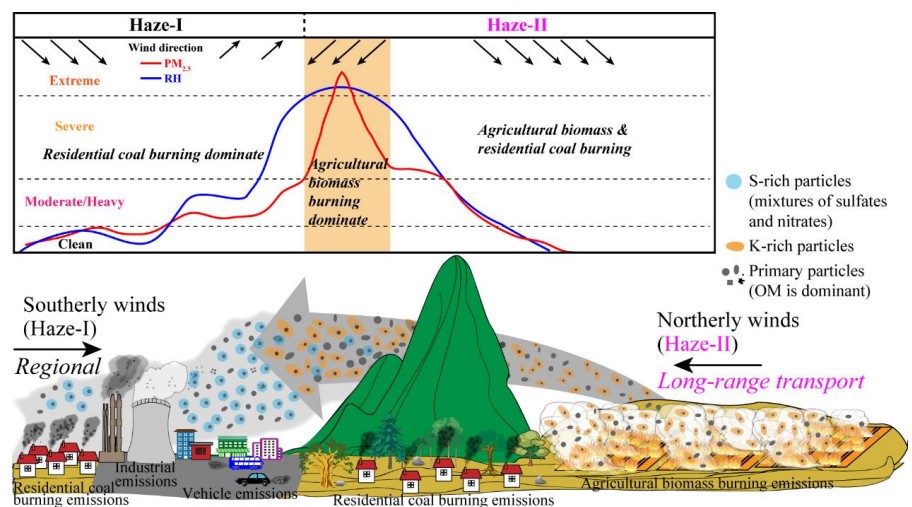

**Figure 8.**