# Peer review of "Exploring wintertime regional haze in Northeast China: role"

_Atmospheric Chemistry and Physics, 2019_

## Referee Comment (RC1) · Anonymous Referee #2 · 30 Jan 2020

General comment

This manuscript by Zhang et al reports morphology, size, composition, aging process, and sources of aerosol particles from regional hazes in Northeast China, where is heavily polluted and thus globally important area. They classified the haze period into haze-I and II and showed differences in the particle compositions. They focus on individual-particle techniques such as TEM, nano-SIMS, and AFM. The result will be important to understand the haze events in the area and possibly the areas located downwind. The results will contribute to the understanding of aerosol particles in the area or other heavily polluted regions. On the other hand, in the discussion, I would argue another possibility for the particle source in the Haze-I, in addition to the possible local source that the authors discuss in this manuscript. The 24-hour back-trajectories

in Fig-S4 show possible contributions from the Beijing area during the Haze-I (heavy) period. The airmass with Beijing pollutants seems to be brought by a low-pressor system on November 4-5 (Fig. S3). Then the airmass changed to Northwest with biomass burning pollutants. If this hypothesis is reasonable, the discussion that the changes in particle composition were due to regional atmospheric chemical reactions (line 427-429) needs to be revised. I suggest to consider the possibility and revised the discussion if needed.

Specific comments

Line 32 and 36 (S-OM and K-OM): Spell out.

Line 43-45: grammatically something wrong.

Line 245: Define DT and NT.

Line 364-367: Please see my general comment. It seems to me that the composition change is mainly due to the airmass change rather than a process of secondary sulfate and nitrate formations.

Line 391-392: Please indicate spherical OM and domelike OM in the figure 6 c-d. I found these OM particles in the clean day in figure 6b but not in figure 6 c-d.

Line 405-410 (figure 7 and S4): Are the back trajectories in Fig. 7 and S4 the same? The Fig. 7 suggests local source but the S4 shows long-range transportation (possibly from Beijing area?). Figure S2 also suggests broader regional pollution events on November 4 rather than a local event. Please also see my general comment.

Line 427-429 and line 570-572: This statement also needs to be reconsidered if my general comment is reasonable.

Line 486: This section includes not only Atmospheric implications but also broader discussion such as health issues. Please reconsider the title.

Line 503-526: I do not see spherical and domelike OM particles in Figure 6c-f when

biomass-burning or coal burning were the dominant sources. Please indicate which particles are spherical and domelike OM particles. When were the particles in Fig 3 collected? The figure shows spherical and domelike OM particles without sampling periods.

Figure 5: Please indicate the Y-axis label on the left.

Table S1: please indicate the error range (or standard deviation).

---

## Referee Comment (RC2) · Anonymous Referee #1 · 8 Mar 2020

This study attempted to integrated bulk chemical measurements with single particle analysis from transmission electron microscopy (TEM), nanoscale secondary ion mass spectrometer (NanoSIMS), and atomic force microscopy (AFM) to obtain morphology, size, composition, aging process, and sources of aerosol particles collected during two contrasting regional haze events (Haze-I and Haze-II) at an urban site and a mountain site in Northeast China. And they also investigated the causes of regional haze formation. Generally, the method was new and sound, and the study showed two haze events and provided information about haze formation in this region. Therefore, this MS may be considered for a publication in ACP after the authors address the following comments.

General comment

[Figure]

I have a major concern about the sampling. The authors should explain whether these 2 haze types within such a short period (1 week) could represent the typical haze type in NE China. What is the sampling strategy behind? What are possible limitations? I suggest the author include non-haze periods for comparisons. Without the detailed discussion, one may not agree that these 2 haze types could represent regional haze formation in NE China. In addition, only weak evidence was present to explain the formation mechanism. This should be clarified carefully.

Abstract: Lines 20-21: this may be not true. Line 23: delete the expression "for the first time" in the abstract.

Methods: PM2.5 mass: quartz filter was not a good option. So this should be compared with the nearby monitoring stations.

Line 259-261: the haze type was not only defined by the wind direction. In addition to the regional transport, changes in emissions and secondary formation play important roles. For example, biomass burning emissions can increase PM2.5 rapidly.

Line 273: what is the major difference for the chemical composition in type 1 and 2. Line 361: so what is the reason of this accumulation? Lines 365-367: are those secondary components formed locally or transported the sampling site? Line 371: what is the reason for such a conversion? Lines 393-394: how to exclude other emission sources? Line 395: coal combustion may emit K-OM particles. Lines 398-399: no direct evidence. How to exclude other sources, e.g., dust, soil, traffic, secondary formation? Lines 409-410: what is the direct evidence? Lines: 427-429: the evidence of heterogeneous reactions should be provided; otherwise it is too speculative.
* * *

---

## Author Comment (AC1) · 18 Mar 2020

**Exploring wintertime regional haze in Northeast China: role of coal and biomass burning**

**Zhang et al.,**

We are grateful for Referee#2's comments. These comments are all valuable and helpful for improving our paper. We answered the comments carefully and have made corrections in the submitted manuscript. The corrections and the responses are as following:

In the revised manuscript and supplement, the red color was marked as the revised places.

**General comment**

1. On the other hand, in the discussion, I would argue another possibility for the particle source in the Haze-I, in addition to the possible local source that the authors discuss in this manuscript. The 24-hour back-trajectories in Fig-S4 show possible contributions from the Beijing area during the Haze-I (heavy) period. The airmass with Beijing pollutants seems to be brought by a low-pressor system on November 4-5 (Fig. S3). Then the airmass changed to Northwest with biomass burning pollutants. If this hypothesis is reasonable, the discussion that the changes in particle composition were due to regional atmospheric chemical reactions (line 427-429) needs to be revised. I suggest to consider the possibility and revised the discussion if needed.

Reply: We appreciated the reviewer' comments and carefully considered them. We added three figures in Figure S5, which shows air mass backward trajectories during the haze period and concentration-weighted trajectory (CWT) plots of PM2.5 on 50 m and 500 m heights during Haze-I event. Figure S5b shows that the contribution of PM2.5 was high from Shenyan city and its surrounding areas and rather low from other regions during the moderate Haze-I period. Figures S5c-d show that compared with transport from Beijing-Tianjin-Hebei region, local emission and transformation mainly contributed to the heavy Haze-I formation. Therefore, we could determine that PM2.5 at the urban and mountain sites were mainly from local emission and transformation during the Haze-I event based on the CWT plots. The detailed discussion as follows.

P17 L425-431: "Through the meteorological data (Figures S4a-b), air mass backward trajectory (Figure 7c), and concentration-weighted trajectory (CWT) plots of PM2.5

(Figures S5b-d) analyses, we inferred that the air quality at the two sampling sites during the Haze-I event was mostly influenced by Shenyang city and its nearby surrounding emissions, although transport from Beijing-Tianjin-Hebei region slightly contributed to the heavy Haze-I formation on 4 DT November."

P17 L436-438: "Figures S5c-d further indicate that compared with transport from Beijing-Tianjin-Hebei region, local secondary transformation was one major factor to cause the heavy Haze-I formation."

**Figure S5.** (a) 24-h air mass backward trajectories on 500 m height before arriving at Shenyang during 31 October-5 November. Concentration-weighted trajectory (CWT) plots of PM2.5 at the urban and mountain sites during the Haze-I event: (b) moderate Haze-I on 500 m height; (c-d) heavy Haze-I on 50 m and 500 m heights.

**Specific comments**

2. Line 32 and 36 (S-OM and K-OM): Spell out.

Reply: We explained S-OM and K-OM particles as follows.

P2 L32-33: "S-rich internally mixed with OM (named as S-OM) particles"

P2 L38: "K-rich internally mixed with OM (named as K-OM) particles"

**3.** Line 43-45: grammatically something wrong.

**Reply: We revised this sentence as follows.**

P2-3 L45-48: "Our study highlights that large numbers of light-absorbing tarballs significantly contribute to winter haze formation in Northeast China and they should be further considered in climate models."

4. Line 245: Define DT and NT.

Reply: We thanked the reviewer. We defined DT and NT in section 2.1.

P7 L154-156: "To better explore the variation of PM2.5 composition and individual particles, we collected the daytime (DT, 8:30-20:00 (local time)) PM2.5 and nighttime (NT, 20:30-8:00 (next day)) PM2.5..."

**5.** Line 364-367: Please see my general comment. It seems to me that the composition change is mainly due to the airmass change rather than a process of secondary sulfate and nitrate formations.

Reply: We added figures of CWT plots of  $PM_{2.5}$  in Figure S5 to indicate the geographic origins of haze particles. Figures S5c-d show that compared with transport from Beijing-Tianjin-Hebei region, local emission and transformation mainly contributed to the heavy Haze-I formation.

We revised this sentence as follows.

P15 L375-378: "Figures 5a-b show that  $SO_4^{2-}/EC$  and  $NO_3^{-}/EC$  factors reached their maximum values at the two sampling sites during the heavy Haze-I event, suggesting that secondary sulfates and nitrates significantly increased during Haze-I evolution (details in Section 4.1)."

**6.** Line 391-392: Please indicate spherical OM and domelike OM in the figure 6 c-d. I found these OM particles in the clean day in figure 6b but not in figure 6 c-d.

Reply: We carefully considered the comment. To make the figure clearer, we indicated spherical OM and domelike OM with red arrows and green arrows in Figures 6a-f as

---

## Author Comment (AC2) · 18 Mar 2020

**Exploring wintertime regional haze in Northeast China: role of coal and biomass burning**

**Zhang et al.,**

We are grateful for Referee#1's comments. These comments are all valuable and helpful for improving our paper. We answered the comments carefully and have made corrections in the submitted manuscript. The corrections and the responses are as following:

In the revised manuscript and supplement, the red color was marked as the revised places.

**General comment**

1. I have a major concern about the sampling. The authors should explain whether these 2 haze types within such a short period (1 week) could represent the typical haze type in NE China. What is the sampling strategy behind? What are possible limitations? I suggest the author include non-haze periods for comparisons. Without the detailed discussion, one may not agree that these 2 haze types could represent regional haze formation in NE China. In addition, only weak evidence was present to explain the formation mechanism. This should be clarified carefully.

Reply: We thanked the reviewer and carefully considered the comments. Firstly, we added the comparisons on chemical compositions during the haze period and non-haze period (clean day) (Figure 2). Here we selected one urban site and one mountain site that cover the regional scale in Northeast China. In fact, we conducted the field campaign about two weeks instead of one week. Here we only selected these two typical haze events as the case. As the referee's concern, we added another one week data in the supplemental materials (Figure S2). But in the main context, we still focused on these two haze events because the data is interesting. In the discussion section, we compared our results with previous studies through radar observation and model simulation, indicating the representativeness of these two haze events.

We added three figures of concentration-weighted trajectory (CWT) plots of PM2.5 in Figures S5b-d, which indicated the geographic origins of haze pollutants. We also added O3 concentrations in Figure S6d, which showed quite low levels of O3 during the heavy Haze-I event. Moreover, we added some marks in Figure 6 to indicate the phases and mixing states of haze particles. These results well contributed to explain the haze formation mechanism.

P6 L130-132: "In this study, we conducted a field experiment in the south part of Northeast China from 25 October to 6 November 2016. Two contrasting regional heavy haze events occurred during 31 October-5 November 2016."

P11 L250-251: "The concentrations of PM2.5 and its chemical compositions during the sampling period were showed in Figure S2."

P11-12 L269-274: "The average mass concentrations of OM, EC, and secondary inorganic ions (i.e.,  $SO_4^{2-}$ ,  $NO_3^{-}$ , and  $NH_4^+$ ) in PM2.5 increased from 11 µg/m3, 3.0 µg/m3, and 6 µg/m3 during the clean day to 45 µg/m3, 4.3 µg/m3, and 24 µg/m3 during the moderate Haze-I event at the urban site and from 9 µg/m3, 0.9 µg/m3, and 3 µg/m3 to 39 µg/m3, 2.7 µg/m3, and 20 µg/m3 at the mountain site, respectively (Figures 2b-c)."

P17 L436-438: "Figures S5c-d further indicate that compared with transport from Beijing-Tianjin-Hebei region, local secondary transformation was one major factor to cause the heavy Haze-I formation."

P18 L441-449: "Because of low O3 level at 12 ppb (Figure S6d) and thick haze layer weakening solar radiation (Zhao et al., 2013; Zheng et al., 2015) during the heavy Haze-I event, photochemical activity should be ignored in air and heterogeneous chemical reactions on particle surfaces can be considered as major pathways in the formation of sulfates and nitrates from SO2 and NOx whenever RH exceeds 70% (Sun et al., 2018; Wang et al., 2016a; Wu et al., 2018). Here, TEM observations did show that the heavy Haze-I particles were wet aerosols and presented S-rich coating on primary OM, soot, or fly ash/metal particles at 73-80% of RH (Figures 6c-d and S4c-d)."

P19 L483-492: "The above formation mechanisms of two haze events are similar to previous studies through radar observation (Zhong et al., 2019) and model simulation (Yang et al., 2017). These studies indicated that adverse meteorological conditions (e.g., low wind speed and high RH) and high-intensity emissions of residential heating and biomass burning mainly contributed to the haze formation in Northeast China in winter. In addition, our previous study suggested that heterogeneous reactions were one major factor for regional winter haze evolution in Northeast China (Zhang et al., 2017). Therefore, we confirmed these two continuous haze events as one typical haze pollution occurring in Northeast China during winter."

**Figure 2.** Variation in the concentrations of  $PM_{2.5}$ , organic matter (OM), elemental carbon (EC), and water-soluble ions (i.e.,  $Ca^{2+}$ ,  $Mg^{2+}$ ,  $K^+$ ,  $Na^+$ ,  $NH_4^+$ ,  $NO_3^-$ ,  $SO_4^{2-}$ ,  $Cl^-$ , and  $F^-$ ) at the urban and mountain sites from daytime (DT) on 31 October to nighttime (NT) on 5 November 2016: (a)  $PM_{2.5}$ ; (b-c) OM, EC, and water-soluble ions. Two regional haze episodes (Haze-I: 31 NT October-4 DT November; Haze-II: 4 NT-5 NT November) were identified.